# Endoscopic Pilonidal Sinus Treatment: Preliminary Results, Learning Curve and Comparison with Standard Open Approach

**DOI:** 10.3390/children10061063

**Published:** 2023-06-15

**Authors:** Giovanni Parente, Francesca Ruspi, Eduje Thomas, Marco Di Mitri, Sara Maria Cravano, Simone D’Antonio, Tommaso Gargano, Mario Lima

**Affiliations:** Pediatric Surgery Department, IRCCS Sant’Orsola-Malpighi Polyclinic, Alma Mater Studiorum—University of Bologna, 40126 Bologna, Italy; francesca.ruspi@studio.unibo.it (F.R.); edu.thomas92@gmail.com (E.T.); marcodimitri14@gmail.com (M.D.M.); sara-cravano@libero.it (S.M.C.); alcmeone1@libero.it (S.D.); tommaso.gargano2@unibo.it (T.G.); mario.lima@unibo.it (M.L.)

**Keywords:** pilonidal cyst, pilonidal sinus, pilonidal disease, endoscopic pilonidal sinus treatment, EPSiT, minimally invasive surgery

## Abstract

**Background:** Pilonidal sinus (PNS) is a common disease which can lead to complications including infection and abscess formation. Different surgical approaches have been suggested, based on primary or secondary wound closure intention healing or endoscopic treatment (EPSiT). The aim of this study is to verify the superior outcomes of EPSiT, especially in comparison with the traditional open approach, and discuss the operators’ learning curve. **Methods**: A retrospective study was conducted identifying all the patients who underwent surgical treatment for PNS with EPSiT technique between 2019 and 2022 and with open technique between 2002 and 2022. We divided patients in two groups: open procedure (group 1) and EPSiT (group 2). We considered a *p* value < 0.05 as statistically significant. **Results:** The mean operative time was 58.6 ± 23.7 min for group 1 and 42.8 ± 17.4 min for group 2 (*p* < 0.01). The mean hospital stay was 2.6 ± 1.7 days for group 1 patients and of 0.8 ± 0.4 days for group 2 (*p* < 0.01). Complete healing was obtained in 18.7 ± 5.6 days for group 1 and 38.3 ± 23.5 days for group 2 (*p* < 0.01). Recurrences were of 23.4% for group 1 and 5.4% for group 2 (*p* = 0.03). **Conclusions:** EPSiT is a minimally invasive and effective approach for the treatment of pilonidal cyst that can be safely performed in pediatric patients with promising results concerning aesthetic outcome and pain control, and with a rapid and less complicated recovery compared to traditional open procedures.

## 1. Introduction

Pilonidal cyst is a relatively common inflammatory lesion of the soft tissues almost always located in the natal cleft. It is a benign but challenging condition for which the treatment is still discussed and has not yet been standardized [1].

Pilonidal sinus (PNS) affects around 26 people in 100,000, with a M:F ratio of 2–3:1 and often presents a positive familiar history for the same disease [2,3,4,5].

The onset takes place usually between adolescence and young adulthood with an often long and complicated natural history, with recurrences being one of the main issues that patients and doctors must deal with.

Described for the first time in 1833 by Herbert Mayo [6], its etiology has been long discussed and nowadays it is recognized as an acquired condition caused by the insertion of free hair into the subcutaneous tissue of the natal cleft [7,8].

The hair trapped in the subcutaneous tissue determines an inflammatory reaction that results in the creation of the so-called pilonidal sinuses and cysts. The cyst can later become infected due to the presence of bacteria deriving both from the hair bulbs themselves or migrating from outside thanks to the sinuses which represent a communication between the skin and the cyst. The infection usually evolves into a pilonidal abscess. Frequently, the abscess self-drains by forming secondary fistulae, but multiple episodes can occur causing pain and discomfort to patients and interfering with their daily lives.

Risk factors for the development of PNS are increased body hair density, a deep gluteal cleft, BMI equal to or greater than 25 kg/m^2^, sedentary lifestyle or prolonged sitting, local skin trauma and family history of pilonidal disease [9].

Different surgical approaches have been suggested, based either on primary wound closure or secondary intention healing, including use of flaps and sometimes complex reconstructive techniques [1,10,11,12,13]. All these procedures are often marked by difficult recovery due to long healing processes, and by significantly high recurrence rate [11,14,15,16].

Minimally invasive procedures were first described in the 1960s [17], and in 2014 Meinero et al. presented the Endoscopic Pilonidal Sinus Treatment (EPSiT), the first endoscopic approach for PNS disease. EPSiT consists of removing hair and cauterizing the cavity from the inside under direct endoscopic vision. This is performed with a fistuloscope equipped with a working and irrigation channel [18,19]. The technique reported good results in the adult population, and in 2018 its first application to pediatric patients was described by Esposito et al. [20]. Good post-operative pain control, rapid recovery and quick healing of the wound are reported to be among the benefits of EPSiT, as well as excellent aesthetic results compared to the traditional open techniques [21,22].

The aim of this study is to verify the superior outcomes of EPSiT after its introduction in our pediatric surgery department, especially in comparison with the traditional open approach, and to comment on the operators’ learning curve.

## 2. Materials and Methods

EPSiT was introduced in our center in September 2019, and since then it has become the standard approach for patients affected by PNS. A retrospective study was conducted identifying all the patients who underwent surgical treatment for PNS at our center with EPSiT technique between 2019 and 2022 and with open technique between 2002 and 2022. Patients operated on with techniques different from the aforementioned ones were excluded from the study, as well as patients who were not completely healed by the time of the enrolment.

Demographic data, such as age and sex, were recorded. Operative and clinical records were screened in search of operative times, surgeons performing the procedure, and anesthesiologic and post-operative pain management.

To compare the dimensions of the pilonidal cysts, we adopted their major diameter. The latter was obtained through the examination of the cyst made by our pathologists in the open group while it was measured during surgery from inside the cyst in the EPSiT group.

To ensure uniformity in the open population, techniques other than excision and wound closure (extremely rare in our institution) were excluded from the study.

Considering the important differences deriving from the recent EPSiT introduction, we arbitrarily set the follow-up of the open group to 9 months.

### 2.1. Open Technique

After visualization of all the sinus orifices, an incision all around them including even paramedian granulomas when present is made. Full excision of the pathologic tissues is performed until the healthy sacrococcygeal fascia underneath is reached. Sometimes, in case of complex and extended cysts, we adopt methylene blue for a better visualization of the sinuses. Complete hemostasis is achieved using diathermy or sometimes argon plasma coagulation. A drainage (in our institution, we prefer a Penrose one) is placed in the wound cavity and, after passing the subcutaneous tissue, fixed outside the wound. Closure of the deep layer is started by interrupted stitches using absorbable stitch material through the deep subcutaneous fat and fascia elevated on both sides together with the central fascia over the sacrum to decrease the dead space. Finally, stitches are taken to approximate the superficial subcutaneous fat; then, the skin is closed using nylon stitch materials. Rarely, in case of extensive excisions or recurrences, the wound is left open to heal naturally.

### 2.2. EPSiT Technique and Post-Operative Management

The EPSiT procedures were conducted either in general anesthesia (GA) or in spinal anesthesia (SA) plus conscious sedation when asked for by the patient, and the main surgical steps are summarized as follows (Figure 1 and Figure 2):Physical examination and introduction of the fistuloscope (8° lens, 3.3 × 4.7 mm, Karl Storz GmbH, Tuttlingen, Germany) through one of the fistula openings after its cruentation if necessary. In the latter case, we use a 4 mm skin punch (Figure 3).Identification of the main sinus and of possible secondary tracts.Removal of the hairs with endoscopic grasper introduced through the fistuloscope working channel.Brushing of inflammatory and necrotic tissue, obtained with an endoscopic brush.Thermocoagulation of the cavity’s walls with an endoscopic monopolar electrode.

Following Esposito’s variation of the traditional EPSiT technique, saline solution was used to irrigate the operative field throughout the procedure instead of a glycine/mannitol solution [18,23].

Contrary to what is stated in the open approach section, we never used methylene blue to enhance sinus visualization. This is because the advantage of this procedure is the possibility to explore the cavity from the inside and, therefore, nearly all the sinuses are visible and explorable.

Post-operative pain management was obtained with intravenous (IV) acetaminophen (15 mg/kg) on a regular basis or on demand. Patients were discharged at first post-operative day (POD) or, in some cases, on the same day of surgery. Medications and follow-up were conducted on an outpatient basis.

At discharge, different instructions regarding wound care were given to the patients and to the caregivers according to the performed procedure. Until complete healing, the open population was instructed to keep the wound dry and to clean it daily with a bland antiseptic. EPSiT patients were divided in two subgroups: group A included patients operated before 09/2021, medicated with topical antibiotic + steroid and hyaluronic acid-based wound dressings, and group B included patients operated after 09/2021, medicated with Novox©, an oxygen-enriched oil-based gel, who were taught to renew the medication every 48 h. Considering the minimally invasive nature of the procedure, we allowed both the EPSiT groups to take a shower after the seventh POD, covering the wound with a waterproof plaster.

After complete healing, we provided to both the open and EPSiT groups a list of long-term care advice, such as daily accurate hygiene (at least twice a day), posture tips and physical activity endorsement. Moreover, laser epilation was recommended.

### 2.3. Statistical Analyses

Statistical analysis of continuous and discrete variables regarding the study population, the surgical procedure and post-operative course was reported in the form of numeric count and percentages. Mean, standard deviation and range were chosen as indicators of central tendency and dispersion of the continuous variables. To detect statistically important differences, t-test was performed for normally distributed variables (after Shapiro–Wilk test of normality). Lastly, exact Fisher test was used to match post-operative therapies and to compare ordinary admission to day surgery inside the subpopulations of the study (GA vs. SA). A *p*-value < 0.05 was considered statistically significant.

## 3. Results

After a preliminary selection, 100 patients were identified for the open group (group 1) but five were excluded from the study (two asportation with secondary intention healing and three incision and drainage of pilonidal abscesses). Therefore, 95 patients were included in the open group and 39 patients in the EPSiT group. Demographic data are reported in Table 1. The two groups are comparable in terms of age and sex since there is no significant difference between the two (*p*-values respectively 0.59 and 0.78).

All the patients belonging to the EPSiT group were operated by two surgeons of our team: 37 procedures conducted by surgeon A and two procedures conducted by surgeon B under supervision of surgeon A.

In group 1, 26 patients (27.3%) had already been subjected to surgical treatment for PNS: 18 abscess drainages, seven asportations with open techniques and one EPSiT procedure performed in another hospital (mean recurrences per patient: 1.8 ± 0.9, range 1–3). In the EPSiT group (group 2), three patients had already been operated one time each with open techniques (7.6%) (Table 1). Data regarding duration of surgery, anesthesiologic management, post-operative pain management and discharge are reported in Table 1, Table 2 and Table 3; moreover, Figure 4A shows the learning curve for the 39 EPSiT cases.

The mean dimension of the pilonidal sinus was of 27.3 ± 20.3 mm (range: 7–80 mm) for group 1 and of 42.6 ± 18.6 mm (10–80 mm) in group 2 (*p* < 0.01).

In the EPSiT population, we identified two subgroups based on the anesthesiologic regimen (GA vs. SA). In the first group, 29/31 (93.5%) patients were admitted as ordinary hospitalization (hospital stay ≥ 1 day), while two were operated in day surgery. Among the SA patients instead, two of eight (25%) patients were converted to ordinary hospital stay due to organizational issues and six underwent day surgery (*p* < 0.01) (Table 2).

During the post-operative period, pain-relieving therapy was classified as shown in Figure 4B. Group 1 patients were managed with acetaminophen IV on regular scheme (46.2%), NSAIDs IV on regular scheme (32.3%) and opioids regular scheme (21.5%). Group 2 patients were managed with acetaminophen IV on regular scheme (56.4%), acetaminophen IV on demand (41%) and opioids IV regular scheme (2.6%).

In group 2, the comparison between regular and on-demand acetaminophen administration showed that in the first case, four patients requested rescue therapy, while only one patient who received on-demand analgesics requested it (*p* = 0.37). The comparison of patients operated in GA with patients operated in SA resulted in 5/31 (16.1%) rescue requests for the first against 0/8 for the latter (*p* = 0.57) (Table 3).

Complete healing, defined as the last outpatient medication performed, was obtained in 18.7 ± 5.6 days (9–30) for group 1 and 38.3 ± 23.5 days (10–84) for group B (*p* < 0.01).

In group 1, the mean number of medications necessary to achieve complete healing was 5.9 ± 9.1 (1–54) while in group 2, it was 4.9 ± 5.0 (1–28) (Table 1).

As previously stated, EPSiT patients were divided in two subgroups. Group A (28/39) included patients operated before 09/2021 and medicated with topical antibiotic + steroid and hyaluronic acid-based wound dressings, who received 5.6 ± 5.8 (1–28) medications. Group B (11/39) comprised patients operated after 09/2021 and medicated with Novox©, an oxygen-enriched oil-based gel, receiving 3.3 ± 1.7 (1–6) medications. The comparison between group 1 and group 2 showed a *p*-value of 0.47, while for EPSiT A vs. EPSiT B, *p* = 0.03 and for group 1 vs. EPSiT B (current standard of practice), *p* = 0.03.

Lastly, recurrences were analyzed at a previously set follow-up of 9 months for group 1 and at a mean follow-up of 7.9 ± 7.3 (0–38) months for group 2 (17.9% vs. 5.4%, *p* = 0.04).

## 4. Discussion

Historically, pilonidal cyst has been a burdensome condition to deal with, mostly due to the long process of recovery and high rates of recurrence [11,14,15,16]. Different open approaches have been developed since the first case description in 1833, and even some flap elements have been acquired from plastic surgery and modified into a suitable technique. Nevertheless, pilonidal cyst treatment still poses some challenges to physicians, and the best surgical approach is still up for debate.

Since its introduction in 2014, EPSiT has been welcomed as the future of the treatment of pilonidal cysts. The first results reported in the literature confirmed initial impressions, showing the superiority of EPSiT over conventional treatments in terms of patient’s tolerability, better aesthetic results and fewer recurrences [24]. After its first application in the pediatric population, described in 2018 by Esposito et al., an increasing number of pediatric surgical units have adopted EPSiT as the treatment of choice. The results reported in the pediatric population appear to match those outlined in the adult population [21]. After two years from its introduction into our practice, we decided to analyze our preliminary results to match them with the current literature and to enrich the debate.

In our practice, mean operative time was found to be significantly shorter for EPSiT procedure (42.8 min vs. 58.6 min, *p* < 0.01) and the analysis of its variation in time showed interesting results. We observed major discrepancies between the first 10 cases, which we believe were necessary for surgeon A to become familiar with the procedure. Following this initial period, we observed a stabilization and progressive decrease in procedural time around the 15th procedure, related to the surgeon’s increased experience. Therefore, we can infer that these results prove how EPSiT is accessible to a large number of surgeons as only a limited number of procedures is required to achieve adequate competence.

The mean hospital stay went from 2.6 days to 0.8 days (*p* < 0.01) considering the whole EPSiT population. Our predictions for the future are that the procedure will mainly be performed on a day surgery basis, as since the introduction of SA, we experienced a significant change in the ratio of ordinary hospitalizations/day surgeries (*p* < 0.01).

The endoscopic approach appeared to be superior also in terms of pain management. As shown in Figure 4B, acetaminophen was sufficient to control the post-operative pain. We argue that its administration should be on demand because the comparison between patients who received it on a regular scheme with the ones who received it on demand proved the non-inferiority of the second choice, as there was no significant change in the rescue doses administered (*p* = 0.37). Likewise, when patients were matched for GA or SA, no increase in rescue medications was detected, showing that SA is as efficient as GA in pain control (*p* = 0.57) and at the same time allows to operate in day surgery. Our outcome regarding post-operative pain matches with the current literature, which highlights a painless recovery and a shorter work leave for patients who underwent endoscopic treatment [25].

In our experience, open procedures were associated with a shorter healing time (18.7 days vs. 3.3, *p* < 0.01), but further considerations are necessary regarding this result: the comparison was conducted between primary closure and secondary intention healing, which explains the difference in time; moreover, as already explained above, we defined the healing time as the moment at which the last outpatient medication was performed. Therefore, considering that EPSiT patients were followed more meticulously during the recovery to have close feedback on the new technique, we may have incurred an observational bias. Finally, all our patients resumed their regular life earlier than the ones who underwent the open approach because of the absence of stitches, tissues under tension and pain. This should be also considered when discussing recovery time.

The introduction of Novox© medications led to a significant improvement in our standard of practice, as the mean number of medications decreased from 5.6 to 3.6 (*p* = 0.03). When comparing open procedures with our current protocol, the result is even more promising, as we went from a mean of 5.9 to 3.3 medications for patients (*p* = 0.03).

Based on the current literature, EPSiT is recommended for non-complex cases of pilonidal cyst when only one pit can be found, or when all the pits are located on the median line [1]. We compared sinus dimensions of group 1 vs. group 2 patients to demonstrate that large cysts should not be excluded from the EPSiT approach. We found that our patients operated with EPSiT generally had bigger cysts (42.6 vs. 27.3 mm, *p* < 0.01) than the ones treated with open surgeries, and outcomes were by no means affected.

Since recurrences and overall final outcome are related not only to the sinus dimension but also to the amount of orifices present in any patient [26], we decided to incorporate this variable in our analysis, and even in this case we did not find any significant difference (1.27 vs. 1.43, *p* = 0.54): once again, we can reasonably infer that the endoscopic approach should be used as first-line treatment in all patients, and not only in a selected subpopulation.

Moreover, we compared our recurrence rate for open patients with the one of the EPSiT group, finding that recurrences went from 23.4% to 5.4% (*p* = 0.03). We can argue that our result is consistent with the current literature on the pediatric population, as Esposito reports a recurrence rate of 4.6% at a median follow-up of 5 years [21,27]. This result confirms the validity of the endoscopic technique in the definitive treatment of pilonidal disease.

Finally, EPSiT advantages are astonishing in terms of cosmesis making it with no doubt superior to the open approach (Figure 5).

### Limits of the Study

We were not able to compare anesthesiologic management between the two techniques (Table 2); this was because to introduce SA was deliberately chosen by anesthesiologists due to EPSiT’s minimally invasive nature. Thus, a comparison on the latter topic between open and EPSiT groups would have led to an important bias.

EPSiT is a newly introduced technique; thus, we have no more than a mid-term follow-up. Even if data are encouraging, a longer follow-up must be taken into account.

To overcome the bias that may derive from too-marked differences in follow-up, we decided to narrow the follow-up window of the open population to 9 months to make it comparable to the mean EPSiT group’s one.

Even if we know that our preliminary results have to be rediscussed after a long-term follow-up analysis, we believe that performing a preliminary analysis when introducing a new technique is mandatory: in order to eventually interrupt a procedure that may give worse results than a standard one.

The novelty of the technique (and, therefore, our initial inexperience) determined a strict follow-up, different from the open one. Therefore, EPSiT patients were subjected to more medication. This difference in meticulousness may represent a bias when comparing the follow-up of the two groups.

Another limitation is the number of patients included as well as the remarkable difference in numerosity between open and EPSiT groups; multicentric studies are needed to confirm our results.

## 5. Conclusions

EPSiT, according to our preliminary results, is a minimally invasive and effective approach to pilonidal cyst that can be safely performed in pediatric patients with promising results concerning aesthetic outcome, pain control and, most importantly, with a quick and less complicated recovery compared to traditional open procedures.

Reports of long-term outcomes for EPSiT are still limited and will be contributions of great value in the debate concerning pilonidal sinus treatment. However, at the current state of knowledge, we advise that EPSiT should be considered and endorsed in the pediatric population.

## Figures and Tables

**Figure 1 children-10-01063-f001:**
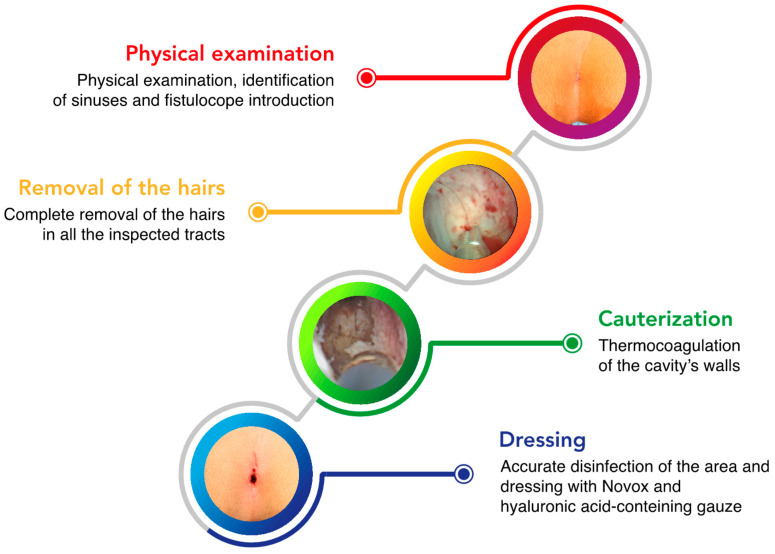
Key passages of EPSiT procedure (see the text for further details).

**Figure 2 children-10-01063-f002:**
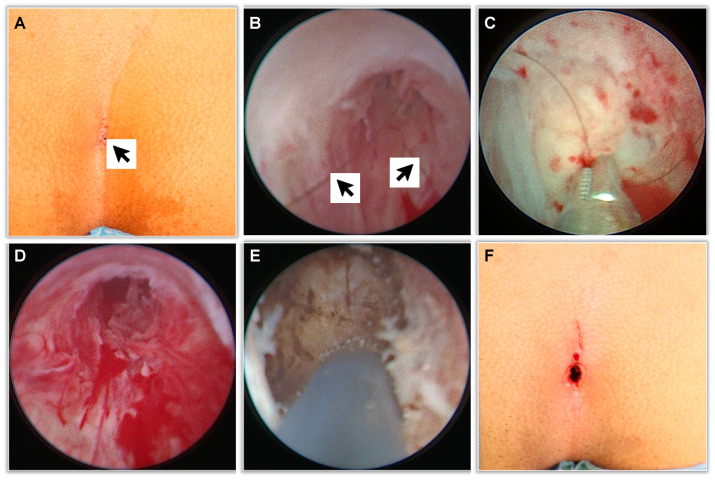
More detailed view of the surgical procedure. (**A**) Physical examination: the arrow shows the sinus opening. (**B**) Endoscopic view at the sinus exploration: the arrows point at the hairs. (**C**) Removal of the hairs with an endoscopic grasper. (**D**) View of the sinus once the hair removal phase is complete. (**E**) Thermocoagulation of the cavity’s wall. (**F**) External appearance at the end of the procedure.

**Figure 3 children-10-01063-f003:**
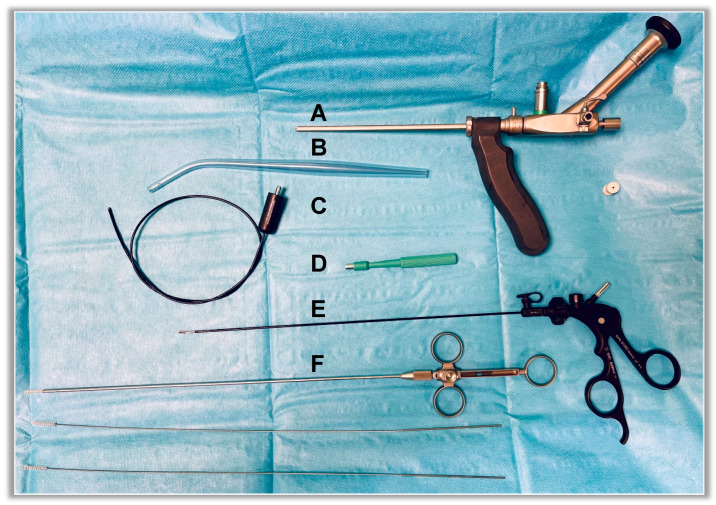
EPSiT main instruments: (**A**) Fistuloscope. (**B**) Yankauer suction tube. (**C**) Monopolar electrode. (**D**) 4 mm biopsy punch. (**E**) Forceps. (**F**) 4 mm, 4.5 mm, and 5 mm fistula brushes.

**Figure 4 children-10-01063-f004:**
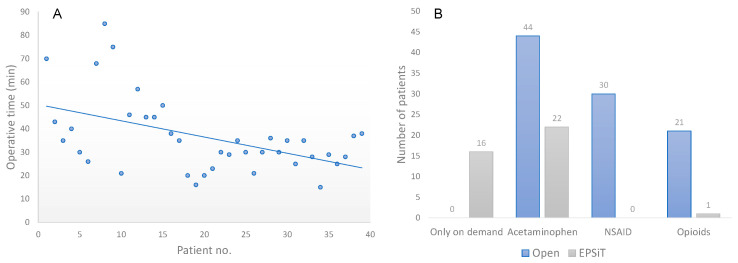
(**A**) EPSiT learning curve. (**B**) Post-operative pain management: comparison between EPSiT and open approach.

**Figure 5 children-10-01063-f005:**
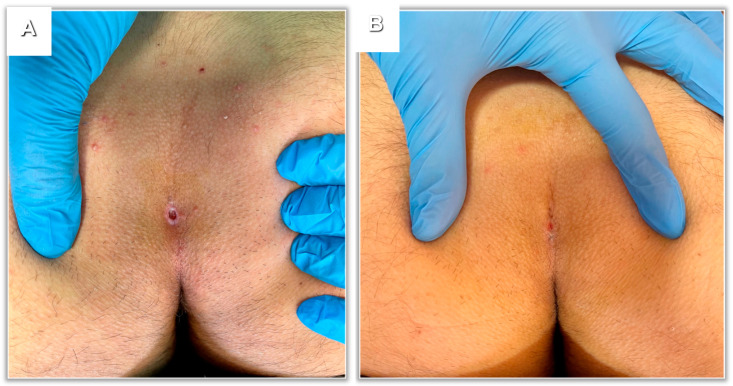
EPSiT post-operative results. (**A**) Physical appearance 7 days after surgery. (**B**) Results 14 days after surgery: wound almost completely healed.

**Table 1 children-10-01063-t001:** Demographic data and results comparison. Continuous data are reported as: mean ± standard deviation (range). N = Number, PNS = Pilonidal Sinus. * Group A is composed of patients who underwent EPSiT procedure before the introduction of Novox in the post-operative medication regimen. ** Group B is composed of patients who underwent EPSiT and received post-operative medication with Novox.

	Open	EPSiT	*p*-Value
N. of patients	95	39	
Male:Female	56:39 (1.5:1)	24:15 (1.6:1)	0.78
Age (years) Less than 16 yo	14.8 ± 2.9 (2–27) 74 (78%)	15.0 ± 2.6 (9–22) 28 (72%)	0.59
Procedural time (min)	58.6 ± 23.7 (13–140)	42.8 ± 17.4 (15–85)	<0.01
PNS dimension (mm)	27.3 ± 20.3 (7–80)	42.6 ± 18.6 (10–80)	<0.01
Hospitalization (days)	2.6 ± 1.7 (0–9)	0.8 ± 0.4 (0–2)	<0.01
Healing time	18.7 ± 5.6 (9–30)	38.3 ± 23.5 (10–84)	<0.01
N. of medications	5.6 ± 5.8 (1–28)	4.9 ± 5.0 (1–28)	0.47
	EPSiT A *	EPSiT B **	
N. of medications	5.6 ± 5.8 (1–28)	3.3 ± 1.7 (1–6)	0.03
	Open	EPSiT B	
N. of medications	5.9 ± 9.1 (1–54)	3.3 ± 1.7 (1–6)	0.03

**Table 2 children-10-01063-t002:** Anesthesia regimen and post-operative pain management data and results comparison.

		Open	EPSiT
Anesthesia regimen	General anesthesia	94/95 (98.9%)	31/39 (79.5%)
	Spinal anesthesia	1/95 (1.1%)	8/39 (20.5%)
Pain management	Acetaminophen	43/93 (46.2%)	22/39 (56.4%)
	Opioids	20/93 (21.5%)	1/39 (2.6%)
	NSAID	30/93 (32.3%)	0/39 (0%)
	On demand Acetaminophen	0/93 (0%)	16/39 (41.0%)

**Table 3 children-10-01063-t003:** Comparison between patients who received scheduled and on-demand pain control therapy and comparison of patients operated under general anesthesia with ones under spinal anesthesia.

	Rescue Analgesics	
	Requested	Not Requested
(A) Patients under scheduled pain-relieving therapy regimen	4/22 (18.2%)	18/22 (81.8%)
(B) Patients under on demand pain-relieving therapy regimen	1/17 (5.9%)	16/17 (94.1%)
(C) Patients who received general anesthesia	5/31 (16.1%)	26/31 (83.9%)
(D) Patients who received spinal anesthesia	0/8 (0%)	8/8 (100%)
	A vs. B	C vs. D
*p*-values	0.37	0.57

## Data Availability

The data presented in this study are available on request from the corresponding author. The data are not publicly available due to privacy restrictions.

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
