# Peer review of "Endoscopic Pilonidal Sinus Treatment: Preliminary Results, Learning Curve and Comparison with Standard Open Approach"

_children, 2023, doi:10.3390/children10061063_

Round 1

Reviewer 1 Report

It is unnecessary to state the abbreviation in the title.

It is unnecessary to mention statistical methods in the abstract. Grouping is also part of the methods, not the results.

Apart from the hair itself, are there other etiological factors closely related to this condition? I suggest you touch on all the factors in the introduction.

"Verify and eventually confirm" are synonyms. I suggest you use one phrase.

To make the manuscript easier to follow, I also suggest that you divide the "materials and methods" section and the "results" section into subsections.

I suggest that in "materials and methods" you briefly describe the open technique that was used in the control group of patients.

In the identification of sinuses and secondary pathways, do you use only visual inspection or, for example, methylene blue?

I suggest you remove figure 1, since figure 2 is representative enough.

Clearly state in the results (expressed in p-values) whether the groups were comparable in terms of age and sex.

In Table 2, clearly state the p-values.

All results, e.g. sinus dimension, in the materials and methods section, the method of measurement must be clearly described.

A number of results are listed both in the text and in the tables. I suggest that what is listed in the tables, do not state again in the text.

Although you have guessed some limitations of your study, there are many more. Spot them all and write them in a separate section at the end of the discussion section.

Also, I suggest that you touch on the discussion section for more articles on this topic, regardless of whether it is a pediatric or adult patient population. Regarding the references from the introduction, the discussion must be enriched with a series of additional references and comparisons on this topic.

In the conclusion, do not state the limitations of the study. The conclusion must be concise, clear and based on its own results.

Moderate correction of the English language is required.

Author Response

Please find attached the answer to your kind comments.

All the added/modified parts according to your suggestions are highlighted in the revised manuscript.

Kind regards.

Reviewer 2 Report

Thank you for submitting your manuscript. Retrospective case series review comparing different surgical approaches for the treatment of the Pilonidal Sinus Disease in a single Institution and reporting their preliminary results of the MIS approach. Several publications support the benefits of the minimal-invasive surgery for the treatment of PNS in the adult and pediatric population.
In general, the manuscript is easy and clear to read.

There are some questions/observations that I would like to propose to the authors in order to improve the content of this manuscript.

Between groups A and B there are many biases and missing data that could affect the statistics and the interpretation of the results such as:

-
One is comparing 20 years vs 3 years of experience in your Institution.
-
In the optic of Pediatrics (as stated in the conclusion), should be important to specify how many patients in group A and B were less than 16 years old?
-
How many surgeons participated in the open approach group in these 20 years? EPSiT has been performed by 1 surgeon in the 95% of the cases
-
why did you decide to include in the open group the 2 patients with the resection and secondary intention healing and the 3 with only incision and drainage, when 95 % of the patients were treated in the same way excision and primary closure”? Do you think this
could affect in someway the statistical results in terms of time of healing, number of controls and recurrence (In the exclusion criteria you informed “patients operated with different techniques were excluded”)

-
the amount of patients operated with recurrences was very different between groups and is missing the information of how many recurrences per patient were registered before the last surgery.
-
Outcome, risk of complications, and recurrences are related to the size of the sinus but also with the amount of orifices associated. Could be possible to include this information in both groups?
-
In the EPSiT group how many patients included the Novox treatment and how many did not?
-
There is an important difference in follow-up between the groups
-
Leaning curve has been reported to be very short. Will be useful to evaluate the increase in performance of EPSiT with other surgeons in your institution, specially that now you have a experienced surgeon in your team that will supervise, maybe much less than 15 cases could
be necessary.

-
In order to be aligned with your conclusions,1) is important to know the amount of pediatric patients included in the study; 2) you are including the aesthetic outcome in EPSiT but in your results you did not evaluate/analyze this issue; 3) you state a rapid recovery but your
results show a longer healing time in the EPSiT group.

Analyzing these important differences between groups and your preliminary results, do you consider to keep in your Title the phrase “..and comparison with Standard Open Approach” or do you prefer to re-evaluate it?

Author Response

(The authors gave the same response as above.)

Round 2

Reviewer 1 Report

Thanks for the answers and for improving the manuscript.

Briefly add the explanation for Table 2 to the limitation of the study.

Again...A number of results are listed both in the text and in the tables. I suggest that what is listed in the tables, do not state again in the text.

Again...Although you have guessed some limitations of your study, there are many more. Spot them all and write them in a separate section at the end of the discussion section.

I suggest that you remove the descriptions from Fig. 5, but add them below in the explanations.

Moderate editing of the English language is required.

Author Response

Please find attached the answers to your kind suggestions.
